# Sleep quality and its predictors among waiters in upscale restaurants: A descriptive study in the Accra Metropolis

**Farrukh Ishaque Saah**[1]ᵒ*, **Hubert Amu**[2]ᵒ

1 Department of Epidemiology and Biostatistics, School of Public Health, University of Health and Allied Sciences, Hohoe, Ghana, 2 Department of Population and Behavioural Sciences, School of Public Health, University of Health and Allied Sciences, Hohoe, Ghana

ᵒ These authors contributed equally to this work.
* fsaahpnur14@uhas.edu.gh

**Data Availability Statement:** Data has been provided as Supporting Information.

**Funding:** The authors received no specific funding for this work.

## Abstract

### Introduction

Poor mental and sleep health negatively affects work performance, turnover intention, and information retention. We examined the impact of waiting job in upscale restaurants on the sleep health of waiters.

### Materials and methods

This was a descriptive study which collected cross-sectional data from waiters of upscale restaurants, using PSQI and DASS-21 questionnaires. Descriptive and inferential statistics comprising mean, frequency, percentage, chi-square, and logistic regression were adopted in presenting the results.

### Results

Sleep quality was poor among 74% of the waiters. The predictors of sleep quality were sex (p = 0.002), role at restaurant (p = 0.004), non-prescription drug use (p<0.001), depression (p<0.001), anxiety (p<0.001), and stress (p<0.001). The prevalence of anxiety, depression, and stress among the waiters was 52.3%, 38.3%, and 34.4% respectively. Stationed (AOR = 4.72, 95%CI = 1.7–812.53, p = 0.002) and supervising (AOR = 3.08, 95%CI = 1.25–7.57, p = 0.014) waiters were more likely to have good sleep quality than headwaiters. Waiters who had depression, anxiety, and stress were, however, 8% (AOR = 0.92, 95%CI = 0.46–1.85, p = 0.819), 28% (AOR = 0.72, 95%CI = 0.38–1.36, p = 0.315), and 49% (AOR = 0.51, 95%CI = 0.24–1.07, p = 0.073) less likely to have a good sleep quality than those who respectively did not have depression, anxiety, and stress.

### Conclusions

Sleep quality was poor among most of the waiters. If this persists, Ghana may not be able to meet the Sustainable Development Goal 3.4 target of promoting mental health and wellbeing. To improve sleep quality and accelerate progress towards achievement of the SDG

**Competing interests:** The authors have declared that no competing interests exist.

target, there should be increased collaboration among stakeholders in the health and hospitality industries to develop innovative interventions to reduce poor sleep quality among workers.

## Introduction

Sleep health is critical to achieving the Sustainable Development Goal (SDG) Three of ensuring healthy lives and promoting wellbeing for all at all ages [1]. To achieve this goal, there is the need to reduce premature mortality from non-communicable diseases through prevention and treatment, and promote mental health and wellbeing, of which sleep quality is critical [1]. Also, inherent in this goal is the "prevention and treatment of substance abuse, including narcotic drug abuse and harmful use of alcohol" [2]. However, waiting work in upscale restaurants has strong impact on the sleep quality and psychological health of waiters. This is because work within the hospitality industry, including restaurants, is labour-intensive and has increasingly harsh environmental demands [3].

Upscale restaurants are designed to appeal to specific group of consumers, and they often offer relatively expensive eatery services [4]. A restaurant is made up of a team of people linked with one another to provide eatery services to customers [5]. Waiters are the frontline employees delivering services in real-time and are, thus, critical to the success of restaurants [6]. Frontline employees are considered the organization's most-central asset and are able to attain and sustain competitive advantage [7]. This is because in a restaurant, customer and employee contact is the first representation of a service, and customer perception is highly influenced.

Working in restaurants can be stressful and the hectic roles pose health risks such as stress for waiters [8]. Usually, this presents as illness, high stress level, life pressure, lack of motivation, work overload [9], 'emotional labour' from long, and anti-social working hours [10]. These are risks for work-related mental health issues such as poor sleep quality, depression, anxiety, and stress [11, 12]. Many challenges related to working in upscale restaurants impact negatively on waiters' job performance and sleep health [13]. For instance, low motivation and poor remuneration [12], hard deadlines, long working hours, night and evening work, repetitive work, high emotional demands, low influence (control), shift work, and problems with coordination of work are very common in upscale restaurants [13, 14]. Also, job demands such as role conflict and role ambiguity cause emotional exhaustion for frontline employees increasing turnover intentions [15].

Sleep health is negatively impacted by bad habits and late working hours (e.g., shift work) in the restaurant sector. Restaurant work also has the likelihood of shift work, alcohol consumption [16] and irregular sleep schedules cause poor sleep quality among service workers, resulting in excessive sleepiness during daytime, insomnia, reduced performance, increased likelihood of work accidents, poor personal relationships, and negative affect (e.g., depression) [17]. In addition, bad habits such as caffeinated consumption likely increase time to fall asleep, reduce sleep hours, and heighten daytime sleepiness [18, 19]. Similarly, alcohol consumption and use of sleep aids such as non-prescription drugs to fall asleep have the potential of interfering with one's normal sleep cycle, reducing sleep quality and daytime alertness [20, 21].

Good sleep quality has been found to increase memory [22] and improve performance [23]. However, few hospitality studies have focused on sleep health [24, 25], and little or no importance has been given to sleep and psychological health of frontline staff in the hospitality industry in most developing countries including Ghana. Despite the importance of waiters as

frontline staff in upscale restaurants in achieving success and maintaining competitive advantage, very limited studies have focused on their health and wellbeing. Also, no study has quantified the prevalence of poor mental health (depression, anxiety, and stress) as well as sleep quality among waiters though most studies have acknowledged that the work of waiters impacts their mental and sleep health. This study, therefore, sought to assess sleep quality and its associated factors among waiters in upscale restaurants in the Accra Metropolis of Ghana. Findings from this study will, thus, fill the knowledge gap as well as present data for evidence-based intervention and policies to improve sleep quality and overall psychosocial health of frontline hospitality workers in Ghana.

## Materials and methods

### Setting

This study was conducted in the Accra Metropolis. The metropolis is one of the administrative districts of the Greater Accra Region and serves as both the capital of the region and the national capital of Ghana [26]. The metropolis shares boundaries with Ga West Municipal in the North, Ga South Municipal to the West, the Gulf of Guinea to the South, and La Dadekotopon Municipal to the East. It stretches over a total land area of 139.674 km$^2$. It has a population of 1,665,086 and represents 42% of the region's total population. Males constitute about 48.1% of the population while 47.0% of the population are migrants [26].

There are about 91.2% of Ghanaians by birth in the metropolis while those who have naturalised constitute 1.3%, with the non-Ghanaian population accounting for 4.0%. The Accra Metropolitan Assembly (AMA) has a high literacy rate of 89%, with more than half (52%) capable of reading and writing in English and other Ghanaian languages [26]. The study specifically took place in six upscale restaurants in Accra Metropolitan. The map of the study setting is shown in Fig 1 [26].

### Study design

The study was a descriptive cross-sectional study adopting a quantitative approach. It was, thus, grounded by the positivist philosophy. The positivist philosophy allowed us to make quantifiable observations leading to statistical analyses [27]. The cross-sectional design allowed for the study of waiters at one point in time, provided the opportunity to select a sample from the waiter population, and made possible generalizations of the sample studied [28].

### Study population and sampling

The study targeted waiters working in upscale restaurants in the Accra Metropolis aged 18 years or older. The study involved waiters working in upscale restaurants who have worked for at least three months at the current facility. However, waiters who were on leave or seriously sick were excluded.

A sample size of 384 was used in this study and it was calculated with Cochran's [29] formula $n = \frac{Z^2 * pq}{d^2}$. The six (6) out of 18 (1/3) upscale restaurants in the metropolis were selected using a balloting approach. This was done by writing the names of all the restaurants on separate pieces of paper, folding them, and putting them in a bowl. A piece of paper was selected at random without replacement after shaking the bowl. This was repeated till all the six restaurants were selected. Nevertheless, a simple random sampling technique was employed in selecting the respondents to participate in the study. Thus, the sample can be considered representative of the larger population. This was done by including at random any waiter at the selected facilities who met the inclusion criteria and accepted to participate in the study.

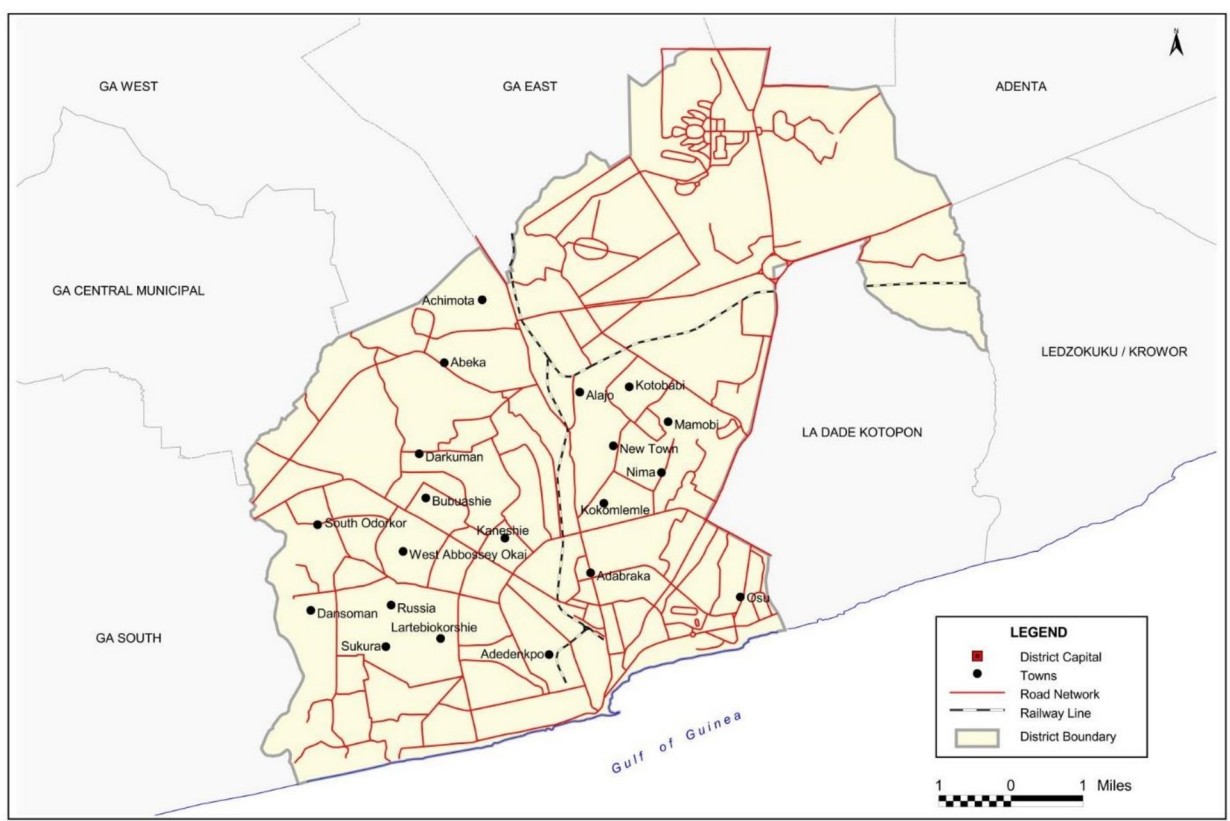

**Fig 1. Map of study setting.**

Waiters were approached during the day's work and those who agreed to participate were included till the sample size was obtained in each restaurant.

## Procedures

A pre-tested questionnaire (S1 Questionnaire) with a Cronbach Alpha of 0.722 was used for the data collection. The questionnaire was pre-tested among waiters working in restaurants in the La Dadekotopon Municipality. The questionnaire was then validated and all shortfalls identified were corrected. The questionnaire comprised socio-demographic form, challenges and prospects associated with upscale restaurants waiting and substance use questions, 21-item Depression Anxiety Stress Scale (DASS-21, Cronbach Alpha = 0.837–0.863) [30], and Pittsburgh Sleep Quality Index Scale (PSQI, Cronbach Alpha = 0.83) [31]. The subscales of DASS-21 had Cronbach's alphas of 0.94, 0.87, and 0.91 for depression, anxiety, and stress, respectively [32]. The DASS-21 is a standard scale for measuring levels of depression, anxiety and stress. It was developed by Lovibond and Lovibond [30] to assess the subjective fundamental symptoms of depression, anxiety, and stress/tension. Also, the PSQI Scale assesses seven components of sleep: subjective sleep quality, sleep latency, habitual sleep efficiency, sleep duration, sleep disturbance, sleep aid medication usage, as well as daytime dysfunction [31].

Data were collected from December 2018 to January 2019 with support from three university graduates (two males and one female) as research assistants. The assistants were trained for two days on the study purpose and the research instruments. Respondents were

approached, the purpose of the study was explained to them, and those who consented were recruited. The questionnaires were self-administered at the respondents' facilities. For respondents who had difficulty in reading and understanding the questions, however, the instruments were researcher-administered. Questionnaires were checked for completeness at close of every data collection session.

### Data analysis

Data were entered into EpiData version 4.1 and exported to Stata 15 for cleaning and analysis. Descriptive and inferential statistics such as means, frequencies and percentages, Pearson chi-square test and binary logistic regression models were conducted. The inferential analysis begun with a chi-square test, followed by significant variables used in the logistic regression analyses. Crude Odds Ratio (COR) and Adjusted Odds Ratio (AOR) were carried out. All statistical analyses were considered significant at $p < 0.05$. The results are presented in tables and charts.

Based on the manual guidelines, scores from each question of the DASS-21 were summed up and multiplied by two to sum suit the original 42-items [30]. Respondents with depression, anxiety, and stress scores of more than 13, 9, and 18 were respectively considered as having depression, anxiety, and stress [30]. The range of scores of each of the seven components of the PSQI was 0 to 3 (0 = no problem). As a result, we computed the Global PSQI Score as the sum of the component scores resulting in a range of 0 to 21. Categorisation of sleep quality was based on its manual and a score of 5 or more was considered poor sleep quality [31]. The dataset for our study is attached as S1 Dataset.

### Ethical issues

Ethical approval for this study was sought from the Ghana Health Service's Research Ethics Committee (GHS-ERC: 63/05/17). Permission was obtained from the managements of the restaurants before data collection. Written consent was obtained from the respondents before including them in the study. This study also ensured the highest level of confidentiality and anonymity in information disclosed to us. This was achieved by ensuring that personal identification information from data collected such as names were removed as well as data collected kept under lock and key without access to a third party. However, respondents who tested positive for any of the conditions were advised to seek professional care.

## Results

### Socio-demographic characteristics of respondents

Table 1 presents the socio-demographic characteristics of the respondents in the study. Of the 384 waiters included in the study, 30.5% were males. Most were 20–24 years old (58.3%), single (70.3%), Christians (83.1%), and had SHS/A'level/O'level of education (72.4%). Almost half of the respondents were Akans (48.2%). Regarding their work, 57.3% of the respondents had worked as waiters for 1–5 years and 54.4% had worked at their current restaurants for 1–5 years. Headwaiters constituted 29.9% while stationed waiters were 63.1%.

### Challenges and prospects associated with waiting in upscale restaurants

Fig 2 shows the challenges encountered by respondents in upscale restaurants. Majority of the respondents experienced job insecurity (59.4%), emotional exhaustion (75.3%), and low motivation (70.3%). Lowered self-esteem and loss of interest were also encountered by 16.9% and 29.9% of the respondents respectively.

**Table 1. Socio-demographic characteristics of respondents.**

| Socio-demographic variable | Frequency | Percentage (%) |
|---|---|---|
| **Sex** | | |
| Male | 117 | 30.5 |
| Female | 267 | 69.5 |
| **Age (in completed years, Mean = 23.03, std. = 3.8)** | | |
| <20 | 55 | 14.3 |
| 20–24 | 224 | 58.3 |
| 25–29 | 87 | 22.7 |
| 30+ | 18 | 4.7 |
| **Marital status** | | |
| Single | 270 | 70.3 |
| Married | 114 | 29.7 |
| **Religion** | | |
| Christian | 319 | 83.1 |
| Muslim | 65 | 16.9 |
| **Highest educational level** | | |
| JHS/JSS | 27 | 7.0 |
| SHS/SSS/A'level/O'level | 278 | 72.4 |
| Tertiary | 79 | 20.6 |
| **Ethnicity** | | |
| Akan | 185 | 48.2 |
| Mole-Dagbani | 36 | 9.4 |
| Ewe | 74 | 19.3 |
| Ga/Dangme | 62 | 16.1 |
| Other | 27 | 7.0 |
| **Years working as a waiter** | | |
| <1 year | 119 | 31.0 |
| 1–5 years | 220 | 57.3 |
| 6–10 years | 37 | 9.6 |
| >10 years | 8 | 2.1 |
| **Years working in current facility** | | |
| <1 year | 163 | 42.4 |
| 1–5 years | 209 | 54.4 |
| 6–10 years | 10 | 2.6 |
| >10 years | 2 | 0.5 |
| **Role at restaurant** | | |
| Headwaiter | 115 | 29.9 |
| Stationed waiter | 242 | 63.1 |
| Supervisor | 27 | 7.0 |
| **Ever been diagnosed with a sleep problem** | | |
| No | 373 | 97.1 |
| Yes | 11 | 2.9 |
| **Sleep problem diagnosed** | | |
| Insomnia | 10 | 90.9 |
| Sleep paralysis | 1 | 9.1 |

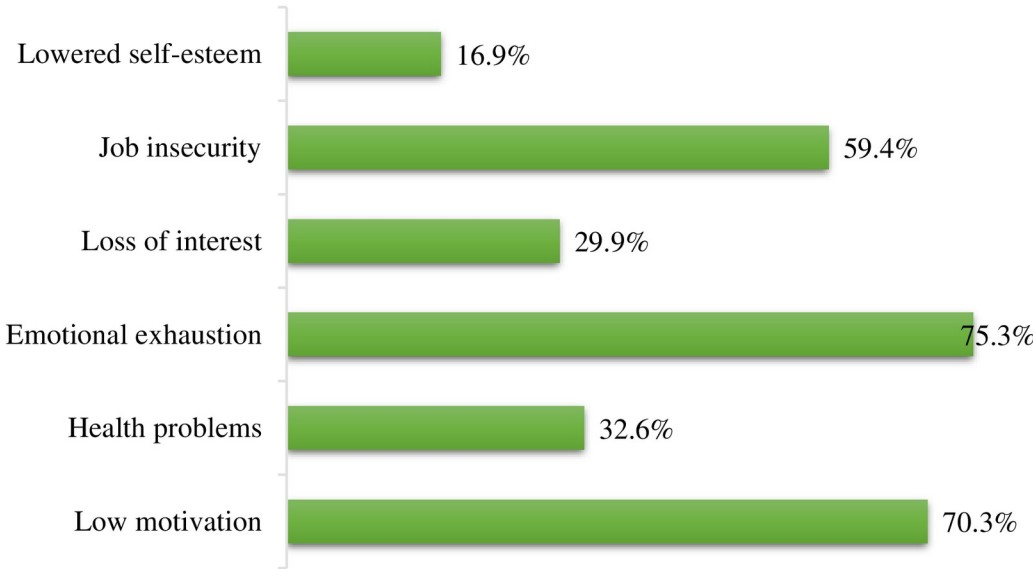

**Fig 2. Challenges associated with waiting job in upscale restaurants.**

From Table 2, most of the respondents were positive about career success (69.3%) and anticipated getting advantage for higher roles/position (68.8%) in current restaurant. However, less than half foresaw the potential of extended work involvement (44.0%) and better remuneration (36.5%).

### Prevalence of depression, anxiety, stress and substance use among waiters in upscale restaurants

Table 3 shows that the prevalence of caffeine and alcohol consumption were 46.6% and 19.3% respectively while cigarette smoking and marijuana use were 2.9% and 1.6% respectively. Also, 43.2% of the respondents used non-prescription drugs. In addition, 38.3%, 52.3%, and 34.4% of the respondents respectively had depression, anxiety, and stress.

**Table 2. Prospects associated with waiting in upscale restaurants.**

| Variable | Frequency | Percentage (%) |
|---|---|---|
| **Positive about career success in current facility** | | |
| No | 118 | 30.7 |
| Yes | 266 | 69.3 |
| **Potential of extended work involvement with current facility** | | |
| No | 215 | 56.0 |
| Yes | 169 | 44.0 |
| **Foresee better remuneration** | | |
| No | 244 | 63.5 |
| Yes | 140 | 36.5 |
| **Anticipate getting an advantage for higher roles/position in current facility** | | |
| No | 120 | 31.3 |
| Yes | 264 | 68.8 |

**Table 3. Depression, anxiety, stress, and substance use among waiters in upscale restaurants.**

| Variable | Frequency | Percentage (%) |
|---|---|---|
| **Caffeine consumption** | | |
| No | 205 | 53.4 |
| Yes | 179 | 46.6 |
| **Alcohol consumption** | | |
| No | 310 | 80.7 |
| Yes | 74 | 19.3 |
| **Cigarette smoking** | | |
| No | 373 | 97.1 |
| Yes | 11 | 2.9 |
| **Marijuana use** | | |
| No | 378 | 99.0 |
| Yes | 6 | 1.6 |
| **Non-prescription drug use** | | |
| No | 218 | 56.8 |
| Yes | 166 | 43.2 |
| **Psychological conditions** | | |
| Depression | 147 | 38.3 |
| Anxiety | 200 | 52.3 |
| Stress | 132 | 34.4 |

## Sleep quality among waiters in upscale restaurant

Fig 3 shows that sleep quality was poor among 74% of the respondents.

## Predictors of sleep quality among waiters in upscale restaurants

Table 4 presents the predictors of sleep quality among waiters in upscale restaurants. Sex (p = 0.002), role at restaurant (p = 0.004), non-prescription drug use (p<0.001), depression level (p<0.001), anxiety level (p<0.001), and stress level (p<0.001) were the predictors of sleep quality among the waiters. Female waiters were, for instance, 48% (AOR = 0.52, 95% CI = 0.31–0.87, p = 0.013) less likely to have good sleep quality than male waiters. Stationed waiters and supervising waiters were 4.72 times (95%CI = 1.7–812.53, p = 0.002) and 3.08 times (95%CI = 1.25–7.57, p = 0.014) respectively more likely to have good sleep quality than headwaiters. Waiters who used non-prescription drugs were 76% (AOR = 0.24, 95%CI = 0.13–0.44, p<0.001) less likely to have good sleep quality. Waiters who had depression, anxiety, and stress were also 8% (AOR = 0.92, 95%CI = 0.46–1.85), 28% (AOR = 0.72, 95%CI = 0.38–1.36), and 49% (AOR = 0.51, 95%CI = 0.24–1.07) less likely to have good sleep quality than those who respectively did not have depression, anxiety, and stress.

## Discussion

This study examined the impact of waiting job in upscale restaurant on sleep health of waiters in Accra, Ghana. We found that challenges faced by the waiters in upscale restaurants included job insecurity, emotional exhaustion, low motivation, lowered self-esteem, and loss of interest. These challenges are similar to those identified in previous studies [9, 13]. These findings may be attributed to existing harsh environmental demands and absence of some intrinsic factors such as managerial support, empowerment, workload, and rewards for work done, which have been reported by other studies [8, 15].

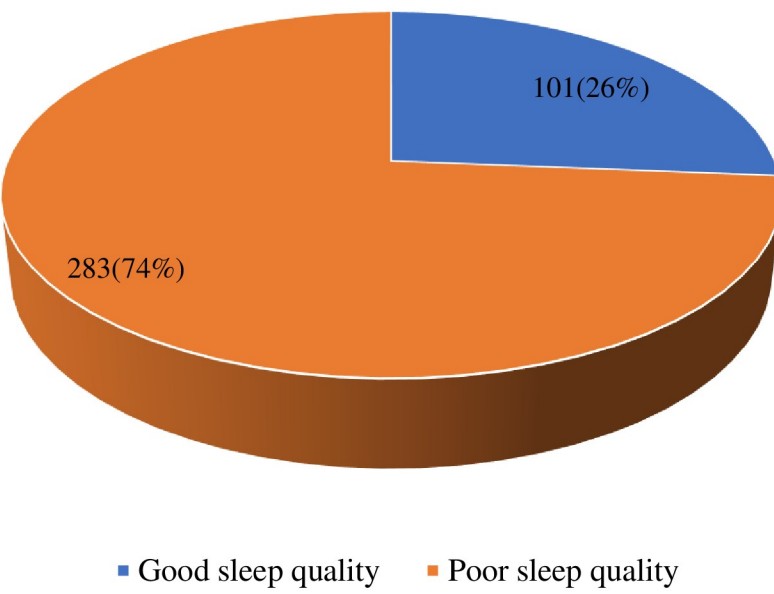

**Fig 3. Sleep quality among waiters in upscale restaurant.**

Nevertheless, positive career success and advantage for higher roles/position were the most observed prospects. However, few waiters foresaw potential of extended work involvement and better remuneration at their place of work. This may be due to increasing attention for how services are produced (service providers) rather than the usual how services are provided [33]. In other words, more emphasis is now being placed on the "people" and "work" extents of most hospitality firms. The prospects found in our study are congruent with those argued to be significant concerns for workers in the hospitality industry [33, 34].

We found that more than a third of the waiters were depressed and stressed while most of them had anxiety. These rates are higher than those reported among other populations. For instance, Maideen and colleagues reported an anxiety rate of 8.2% in a Malaysian study [35]. The prevalence of depression observed in our study is also comparatively higher than the 18.7% reported in another study [36]. However, the stress rate is lower than the 39% found in a previous study [37]. These findings confirm the argument that work-related mental health issues such as depression, anxiety, and stress are common among hospitality staff including waiters [11, 12]. Our findings point to the hectic roles, long and anti-social work schedules, and poor working conditions of waiters in upscale restaurants [10]. The stressful nature of the work of waiters together with challenges and limited prospects at current facility could be a source of strain on their mental health.

We found that many of the waiters consumed caffeine, alcohol, and non-prescription drugs. These findings support previous studies that ranked the food service industry as top-most industry for employee substance use [38, 39]. A possible reason for the high prevalence of caffeine, alcohol, and non-prescription drugs use could be the need to stay awake and active to carry out responsibilities, aiding sleep, and the availability of alcohol and caffeine at work place [40].

We found that most of the waiters had poor sleep quality. The prevalence recorded in our study is higher than those reported by Ghalichi et al. and Yang et al. [41, 42]. The high prevalence of poor sleep may be due to the long hours and shift work associated with working in upscale restaurants [43, 44]. This is because shift work presents many problems associated

**Table 4. Predictors of sleep quality among waiters in upscale restaurants.**

| Variable | Sleep quality | | $\chi^2$(p-value) | COR(95%CI)p-value | AOR(95%CI)p-value |
|---|---|---|---|---|---|
| | Poor n(%) | Good n(%) | | | |
| **Sex** | | | 9.48(0.002)** | | |
| Male | 74(63.2) | 43(36.8) | | Ref | Ref |
| Female | 209(78.3) | 58(21.7) | | 0.48(0.30–0.77)0.002** | 0.52(0.31–0.87)0.013* |
| **Age (in completed years)** | | | 3.14(0.371) | | |
| <20 | 43(78.2) | 12(21.8) | | | |
| 20–24 | 168(75.0) | 56(25.0) | | | |
| 25–29 | 58(66.7) | 29(33.3) | | | |
| 30+ | 14(77.8) | 4(22.2) | | | |
| **Marital status** | | | 2.33(0.127) | | |
| Single | 205(75.9) | 65(24.1) | | | |
| Married | 78(68.4) | 36(31.6) | | | |
| **Religion** | | | 0.35(0.556) | | |
| Christian | 237(74.3) | 82(25.7) | | | |
| Muslim | 8(12.3) | 19(29.2) | | | |
| **Highest educational level** | | | 3.67(0.158) | | |
| JHS/JSS | 22(81.5) | 5(18.5) | | | |
| SHS/SSS/A'level/O'level | 209(75.2) | 69(24.8) | | | |
| Tertiary | 52(65.8) | 27(34.2) | | | |
| **Ethnicity** | | | 1.97(0.742) | | |
| Akan | 138(74.6) | 47(25.4) | | | |
| Mole-Dagbani | 28(77.8) | 8(22.2) | | | |
| Ewe | 50(67.6) | 24(32.4) | | | |
| Ga-Dangme | 47(75.8) | 15(24.2) | | | |
| Other | 20(74.1) | 7(25.9) | | | |
| **Years working as a waiter** | | | 1.67(0.644) | | |
| < 1 year | 91(76.5) | 28(23.5) | | | |
| 1–5 years | 158(71.8) | 62(28.2) | | | |
| 6–10 years | 27(73.0) | 10(27.0) | | | |
| >10 years | 7(87.5) | 1(12.5) | | | |
| **Years working in current facility** | | | 0.84(0.841) | | |
| < 1 year | 121(74.2) | 42(25.8) | | | |
| 1–5 years | 153(73.2) | 56(26.8) | | | |
| 6–10 years | 8(80.0) | 2(20.0) | | | |
| >10 years | 1(50.0) | 1(50.0) | | | |
| **Role at restaurant** | | | 10.85(0.004)** | | |
| Headwaiter | 91(79.1) | 24(20.9) | | Ref | Ref |
| Station waiter | 179(74.0) | 63(26.0) | | 4.08(4.70–9.83)0.002** | 4.72(1.78–12.53)0.002** |
| Supervisor | 13(48.1) | 14(51.9) | | 3.06(1.36–6.86)0.007** | 3.08(1.25–7.57)0.014* |
| **Ever been diagnosed with a sleep problem** | | | 0.01(0.941) | | |
| No | 275(73.7) | 98(26.3) | | | |
| Yes | 8(72.7) | 3(27.3) | | | |
| **Positive of career success** | | | 0.99(0.319) | | |
| No | 83(70.3) | 35(29.7) | | | |
| Yes | 200(75.2) | 66(24.8) | | | |
| **Potential of extended work** | | | 0.02(0.898) | | |
| No | 159(74.0) | 56(26.0) | | | |

*(Continued)*

**Table 4.** (Continued)

| Variable | Sleep quality | | $\chi^2$(p-value) | COR(95%CI)p-value | AOR(95%CI)p-value |
|---|---|---|---|---|---|
| | Poor n(%) | Good n(%) | | | |
| Yes | 124(73.4) | 45(26.6) | | | |
| **Foresee better remuneration** | | | 3.55(0.060) | | |
| No | 172(70.5) | 72(29.5) | | | |
| Yes | 111(79.3) | 29(20.7) | | | |
| **Anticipate an advantage for higher roles/position** | | | 3.46(0.063) | | |
| No | 81(67.5) | 39(32.5) | | | |
| Yes | 202(76.5) | 62(23.5) | | | |
| **Caffeine consumption** | | | 0.51(0.474) | | |
| No | 148(72.2) | 57(27.8) | | | |
| Yes | 135(75.4) | 44(24.6) | | | |
| **Alcohol consumption** | | | 0.025(0.875) | | |
| No | 229(73.9) | 81(26.1) | | | |
| Yes | 54(73.0) | 20(27.0) | | | |
| **Cigarette smoking** | | | 1.73(0.188) | | |
| No | 273(73.2) | 100(26.8) | | | |
| Yes | 10(90.9) | 1(9.1) | | | |
| **Marijuana use** | | | 0.29(0.589) | | |
| No | 278(73.5) | 100(26.5) | | | |
| Yes | 5(83.3) | 1(16.7) | | | |
| **Non-prescription drug use** | | | 36.05(<0.001) | | |
| No | 135(61.9) | 83(38.1) | | Ref | Ref |
| Yes | 148(89.2) | 18(10.8) | | 0.20(0.11–0.35)<0.001 | 0.24(0.13–0.44)<0.001 |
| **Depression level** | | | 12.23(<0.001) | | |
| Normal | 160(67.5) | 77(32.5) | | Ref | Ref |
| Depressed | 123(83.7) | 24(16.3) | | 0.41(0.24–0.68)0.001** | 0.92(0.46–1.85)0.819 |
| **Anxiety level** | | | 15.32(<0.001) | | |
| Normal | 118(64.5) | 65(35.5) | | Ref | Ref |
| Anxious | 165(82.1) | 36(17.9) | | 0.40(0.25–0.63)<0.001 | 0.72(0.38–1.36)0.315 |
| **Stress level** | | | 20.87(<0.001) | | |
| Normal | 167(66.3) | 85(33.7) | | Ref | Ref |
| Stressed | 116(87.9) | 16(12.1) | | 0.27(0.15–0.49)<0.001 | 0.51(0.24–1.07)0.073 |

*p<0.05,

**p<0.01

COR: Crude Odds Ratio AOR: Adjusted Odds Ratio.

with sleep, including insufficient sleep (duration), difficulty in getting to sleep (sleep latency), and feeling unrefreshed after sleep [45]. It, thus, increases the risk of having poor sleep quality. In addition, psychosocial concerns often lead to less quality sleep and inadequate sleep or changed sleep patterns [46, 47].

Predictors of sleep quality among the waiters were sex, role at restaurant, non-prescription drug use, depression, anxiety, and stress. The finding that waiters' sex significantly influenced the risk of poor sleep quality is in congruence with findings from other studies [48, 49]. Similar to these studies, we found that females were more at risk than males. It is likely that due to the home chores and family responsibilities sometimes female waiters perform such as preparing family meal and caring for children, they may be experiencing early sleep time and waking up

and longer sleep latency [49]. Our finding where non-prescription drug use predicted quality of sleep among the waiters elucidates the view that use of non-prescribed drug may affect sleep and subsequently reduce the health-related quality of life [50]. This could possibly be that these non-prescription drugs were self-medicated to treat sleep problems or body aches from stress associated with waiting work [51].

Our findings that depression, anxiety, and stress significantly predicted sleep quality confirm the position by Anxiety and Depression Association of America that mental health problems such as stress and anxiety may result in sleep problems or worsen existing sleep problems [52]. It was, thus, obvious in our regression analyses that waiters who were depressed, anxious, or stressed recorded lower probabilities of good sleep quality. Our study is also, thus, congruent with findings of Brand et al. and Yang et al. that stress, anxiety, and depression are strong correlates of sleep quality [25, 42]. These mental health conditions impact the ability to sleep, the duration it takes for the individual to sleep, and the length of sleep [53]. This situation could likely be exacerbated among waiters as they continue to work stressfully, increasing the effect of the condition on their sleep quality.

## Strengths and limitations

A key strength of this study is our use of standardized assessment tools such as DASS-21 and PSQI in defining cases of depression, anxiety and stress, and sleep quality respectively. Nevertheless, there was the tendency of over-reporting good behaviours, as the study relied on verbal reports of respondents. Efforts were, however, made to explain to the respondents the need to be honest in all their responses. Also, the level of physical activity among the study respondents was not assessed in this study even though it could be an important predictor of sleep quality.

## Conclusions

Waiters in upscale restaurants experience high levels of depression, anxiety, and stress. They consume alcohol, caffeine, or non-prescription drugs as avenues of overcoming fatigue, lack of interest, and sleepiness or a lack of it. Sleep quality is also poor among them. Our findings suggest that current efforts to achieve SDG 3.5 target of preventing and treating substance abuse and harmful use of alcohol by Ghana may be generally inadequate. Ghana may not be able to meet the Sustainable Development Goal 3.4 target of reducing premature mortality from non-communicable diseases and promoting mental health and wellbeing by the year 2030, should the levels of anxiety, depression, and stress as well as poor sleep quality recorded in this study persist. In the long term also, the restaurant industry is likely to lose its workforce, as turnover may increase due to poor health.

## Recommendations

To improve sleep quality and accelerate progress towards achievement of the SDG target, there should be increased collaboration among stakeholders in the health and hospitality industries to develop innovative interventions to reduce poor sleep quality among workers. Interventions should also be implemented to address depression, anxiety, stress and poor sleep quality among staff. The Ghana Health Service in collaboration with international organisations such as the World Health Organisation should increase awareness creation and implement interventions targeted at the prevention of mental health problems. The International Labour Organization, the Ghana Tourism Authority, and the Ghana Tourism Federation should also work to develop policies to improve the overall working conditions and environment of employees in the hospitality industry.

## Supporting information

**S1 Questionnaire. Questionnaire on sleep quality and associated factors among waiters.**
(PDF)

**S1 Dataset. Dataset on assessing sleep quality and associated factors among waiter.**
(SAV)

## Acknowledgments

We acknowledge our research assistants for their support in collecting the data for this study. We are also grateful to Mr. Ebenezer Agbaglo, who copyedited this manuscript.

## Author Contributions

**Conceptualization:** Farrukh Ishaque Saah, Hubert Amu.

**Data curation:** Farrukh Ishaque Saah, Hubert Amu.

**Formal analysis:** Farrukh Ishaque Saah, Hubert Amu.

**Funding acquisition:** Farrukh Ishaque Saah.

**Investigation:** Farrukh Ishaque Saah, Hubert Amu.

**Methodology:** Farrukh Ishaque Saah, Hubert Amu.

**Project administration:** Farrukh Ishaque Saah, Hubert Amu.

**Resources:** Farrukh Ishaque Saah, Hubert Amu.

**Software:** Farrukh Ishaque Saah, Hubert Amu.

**Supervision:** Farrukh Ishaque Saah, Hubert Amu.

**Validation:** Farrukh Ishaque Saah, Hubert Amu.

**Visualization:** Farrukh Ishaque Saah.

**Writing – original draft:** Farrukh Ishaque Saah, Hubert Amu.

**Writing – review & editing:** Farrukh Ishaque Saah, Hubert Amu.

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
