## [Decision Letter · Decision Letter 0]

8 Sep 2020

PONE-D-20-07362

Sleep quality and its predictors among waiters in upscale restaurants: A descriptive study in Accra Metropolis

PLOS ONE

Dear Farrukh Ishaque Saah,

Thank you for submitting your manuscript to PLOS ONE. After careful consideration, we feel that it has merit but does not fully meet PLOS ONE’s publication criteria as it currently stands. Therefore, we invite you to submit a revised version of the manuscript that addresses the points raised during the review process.

The statistical analysis needs specific attention where you are reporting on logistic regression analysis.

We look forward to receiving your revised manuscript.

Kind regards,

Ali Montazeri

Academic Editor

PLOS ONE

Journal Requirements:

2. In your Methods section, please provide additional information about the participant recruitment method and the demographic details of your participants. Please ensure you have provided sufficient details to replicate the analyses such as: a) the recruitment date range (month and year), b) a description of any inclusion/exclusion criteria that were applied to participant recruitment, c) a table of relevant demographic details, d) a statement as to whether your sample can be considered representative of a larger population, e) a description of how participants were recruited, and f) descriptions of where participants were recruited and where the research took place.

3. Please include additional information regarding the survey or questionnaire used in the study and ensure that you have provided sufficient details that others could replicate the analyses. For instance, if you developed a questionnaire as part of this study and it is not under a copyright more restrictive than CC-BY, please include a copy, in both the original language and English, as Supporting Information. Moreover, please include more details on how the questionnaire was pre-tested, and whether it was validated.

4. Please correct your reference to "p=0.000" to "p<0.001" or as similarly appropriate, as p values cannot equal zero.

5. In your Conclusions and Abstract sections, please ensure that only conclusions that can be directly drawn from the data here presented are reported. For example, it is not clear how the data shown in the present analysis can support the statements at lines 268 to 273.

6.We suggest you thoroughly copyedit your manuscript for language usage, spelling, and grammar. If you do not know anyone who can help you do this, you may wish to consider employing a professional scientific editing service.  

7. We note that you have stated that you will provide repository information for your data at acceptance. Should your manuscript be accepted for publication, we will hold it until you provide the relevant accession numbers or DOIs necessary to access your data. If you wish to make changes to your Data Availability statement, please describe these changes in your cover letter and we will update your Data Availability statement to reflect the information you provide.

8.We note that [Figure(s) 1] in your submission contain [map/satellite] images which may be copyrighted. All PLOS content is published under the Creative Commons Attribution License (CC BY 4.0), which means that the manuscript, images, and Supporting Information files will be freely available online, and any third party is permitted to access, download, copy, distribute, and use these materials in any way, even commercially, with proper attribution. For these reasons, we cannot publish previously copyrighted maps or satellite images created using proprietary data, such as Google software (Google Maps, Street View, and Earth). For more information, see our copyright guidelines: http://journals.plos.org/plosone/s/licenses-and-copyright.

1.    You may seek permission from the original copyright holder of Figure(s) [1] to publish the content specifically under the CC BY 4.0 license. 

Reviewers' comments:

**Comments to the Author**

1. Is the manuscript technically sound, and do the data support the conclusions?

Reviewer #1: Yes

Reviewer #2: Partly

2. Has the statistical analysis been performed appropriately and rigorously? 

Reviewer #1: Yes

Reviewer #2: Yes

3. Have the authors made all data underlying the findings in their manuscript fully available?

Reviewer #1: Yes

Reviewer #2: Yes

4. Is the manuscript presented in an intelligible fashion and written in standard English?

Reviewer #1: Yes

Reviewer #2: Yes

5. Review Comments to the Author

Reviewer #1: The aim of this study was to assess the impact of waiting job in upscale restaurant on the health and well-being of waiters. Additionally, the authors assessed sleep quality and its associated factors among waiters in upscale restaurants. This is a very interesting study and I have just a minor comment related to sleep quality predictors. The authors should add in the Limitations section that also the level of physical activity, which was not assesesd in this study, could also be a predictor of sleep quality.

Reviewer #2: PONE-D-20-07362

The manuscript entitled ‘Sleep quality and its predictors among waiters in upscale restaurants: A descriptive study in Accra Metropolis’ aimed to assess sleep quality and its associated factors among waiters in upscale restaurants in the Accra Metropolis of Ghana’. It seems that the methodology and the result are adequate, but some revisions are required. There are some comments as follow.

- First page of introduction is just about growing fast food restaurants and the importance of competition between restaurant and so on (which is not your main goal). The introduction needs a revision in order to highlight the sleep quality more and the factors affecting that. The reasons why this happens might to summarize in a single paragraph. Please summarize the introduction.

- How this study would help and be of importance? Please mention in the introduction more clearly and in brief.

- The last two sentences of introduction are saying two different things. Mention the overall and primary goal of your study.

- There is a main point regarding sampling method. How did you choose the restaurants (what kind of sampling methods)? The result can generalize to the whole population of waiters if we use probability sampling methods. But it is not clear how you chose restaurants.

- Please mention the full phrases of abbreviations in the last raw of tables. What are COR and AOR in the table 4, for example.

- What statistical analysis is used (the results in Table 4)? Mention in the statistical analysis part (just mentioned in the abstract). As you used logistic regression, so mention odds ratio (OR) in the statistical analysis and also use the abbreviation of OR in the manuscript. Why most of the cells are empty in the Table 4?

6. PLOS authors have the option to publish the peer review history of their article (what does this mean?). If published, this will include your full peer review and any attached files.

Reviewer #1: No

Reviewer #2: No

---

## [Author Response · Author response to Decision Letter 0]

14 Sep 2020

Please find below our step-by-step response to the comments raised:

1. Journal Requirements

Comment 1: Please ensure that your manuscript meets PLOS ONE's style requirements, including those for file naming.

Response: The manuscript has been reformatted to meet PLOS ONE’s style requirements.

Comment 2: In your Methods section, please provide additional information about the participant recruitment method and the demographic details of your participants. Please ensure you have provided sufficient details to replicate the analyses such as: a) the recruitment date range (month and year), b) a description of any inclusion/exclusion criteria that were applied to participant recruitment, c) a table of relevant demographic details, d) a statement as to whether your sample can be considered representative of a larger population, e) a description of how participants were recruited, and f) descriptions of where participants were recruited and where the research took place.

Response: The methods section has been revised appropriately (See page 7 to 8). A table of relevant demographic details is presented as part of the Results section (See page 10).

Comment 3: Please include additional information regarding the survey or questionnaire used in the study and ensure that you have provided sufficient details that others could replicate the analyses. For instance, if you developed a questionnaire as part of this study and it is not under a copyright more restrictive than CC-BY, please include a copy, in both the original language and English, as Supporting Information. Moreover, please include more details on how the questionnaire was pre-tested, and whether it was validated.

Response: Additional information on study questionnaire, pre-testing and validation has been provided (see page 8). The research instrument has also been provided as an additional file.

Comment 4: Please correct your reference to "p=0.000" to "p<0.001" or as similarly appropriate, as p values cannot equal 

zero.

Response: The p-values have been corrected to p<0.001 as recommended by the reviewer (See page 18).

Comment 5: In your Conclusions and Abstract sections, please ensure that only conclusions that can be directly drawn from the data here presented are reported. For example, it is not clear how the data shown in the present analysis can support the statements at lines 268 to 273.

Response: The conclusions in the revised manuscript have been linked to the key findings of the study (See pages 22) as the hospitality industry employs a substantial proportion of the general population.

Comment 6: We suggest you thoroughly copyedit your manuscript for language usage, spelling, and grammar. If you do not know anyone who can help you do this, you may wish to consider employing a professional scientific editing service.

Response: The manuscript has been copyedited by an expert, Mr. Ebenezer Agbaglo who is an English Language expert. He has been duly acknowledged in the revised manuscript (See page 23).

Comment 7: We note that you have stated that you will provide repository information for your data at acceptance. Should your manuscript be accepted for publication, we will hold it until you provide the relevant accession numbers or DOIs necessary to access your data. If you wish to make changes to your Data Availability statement, please describe these changes in your cover letter and we will update your Data Availability statement to reflect the information you provide.

Response: Contrary to the initial data availability statement. We have now provided the data used for our analysis as an additional file. Our data availability has, thus, also been changed to reflect same.

Comment 8: We note that [Figure(s) 1] in your submission contain [map/satellite] images which may be copyrighted. All PLOS content is published under the Creative Commons Attribution License (CC BY 4.0), which means that the manuscript, images, and Supporting Information files will be freely available online, and any third party is permitted to access, download, copy, distribute, and use these materials in any way, even commercially, with proper attribution. For these reasons, we cannot publish previously copyrighted maps or satellite images created using proprietary data, such as Google software (Google Maps, Street View, and Earth).

Response: Figure 1 is not copyrighted. The source publication has, however, been duly cited in our revised manuscript.

2. Reviewer #1

Comment: The authors should add in the Limitations section that also the level of physical activity, which was not assessed in this study, could also be a predictor of sleep quality.

Response: This has been done (See page 21).

3. Reviewer #2

Comment 1: First page of introduction is just about growing fast food restaurants and the importance of competition between restaurant and so on (which is not your main goal). The introduction needs a revision in order to highlight the sleep quality more and the factors affecting that. The reasons why this happens might to summarize in a single paragraph. Please summarize the introduction.

Response: The introduction has been revised as recommended by the reviewer (See pages 4-6).

Comment 2: How this study would help and be of importance? Please mention in the introduction more clearly and in brief.

Response: The importance of this study has been presented in the introduction (See page 6).

Comment 3: The last two sentences of introduction are saying two different things. Mention the overall and primary goal of your study.

Response: The two sentences have been reconciled to present the primary goal of the study (See page 6).

Comment 4: There is a main point regarding sampling method. How did you choose the restaurants (what kind of sampling methods)? The result can generalize to the whole population of waiters if we use probability sampling methods. But it is not clear how you chose restaurants.

Response: The sampling method used in selecting the six restaurants and how it was carried out has been explained in the revised manuscript (See page 7).

Comment 5: Please mention the full phrases of abbreviations in the last raw of tables. What are COR and AOR in the table 4, for example.

Response: The full phrases of abbreviation (COR and AOR) in table 4 has been stated in the last raw of the table (see page 18) as well as in the Analysis sub-section (see page 9).

Comment 6: What statistical analysis is used (the results in Table 4)? Mention in the statistical analysis part (just mentioned in the abstract). As you used logistic regression, so mention odds ratio (OR) in the statistical analysis and also use the abbreviation of OR in the manuscript. Why most of the cells are empty in the Table 4?

Response: The statistical analysis used has been stated. In the logistic regression analyses, only variables that were significant in the Chi square test were used that is why some of the cells are empty. This has been stated in the Analysis sub-section (see page 9).

---

## [Editor Report · Decision Letter 1]

30 Sep 2020

Sleep quality and its predictors among waiters in upscale restaurants: A descriptive study in Accra Metropolis

PONE-D-20-07362R1

Dear Dr. Saah,

We’re pleased to inform you that your manuscript has been judged scientifically suitable for publication and will be formally accepted for publication once it meets all outstanding technical requirements.

Kind regards,

Ali Montazeri

Academic Editor

PLOS ONE
---

## [Editor Report · Acceptance letter]

2 Oct 2020

PONE-D-20-07362R1 

Sleep quality and its predictors among waiters in upscale restaurants: A descriptive study in the Accra Metropolis 

Dear Dr. Saah:

I'm pleased to inform you that your manuscript has been deemed suitable for publication in PLOS ONE. Congratulations! Your manuscript is now with our production department. 

Kind regards, 

on behalf of

Professor Ali Montazeri 

Academic Editor

PLOS ONE